# CC-VFed: Client Contribution detects Byzantine Attacks in Vertical Federated Learning

## Abstract

Vertical federated learning (VFL) is a type of federated learning where the collection of different features is shared among multiple clients, and it is attracting attention as a training method that takes into account the privacy and security of training data. On the other hand, in federated learning, there is a threat of Byzantine attacks, where some malicious clients disrupt the training of the model and output an trained model that does not exhibit the behavior that should be obtained. Thus far, numerous defense methods against Byzantine attacks on horizontal federated learning have been proposed, most of which focus on the similarity of the models generated across clients having the similar features and mitigate the attacks by excluding outliers. However, in VFL, the feature sets assigned by each client are inherently different, making similar methods inapplicable, and there is little existing research in this area. In light of the above, this paper organizes and classifies feasible Byzantine attacks and proposes a new defense method CC-VFed against these attack methods. Firstly, this paper organizes and classifies attack methods that contaminate training data, demonstrating that sign-flipping attacks pose a threat to VFL. Subsequently, in order to capture the differences in client features, this paper proposes a method for detecting and neutralizing malicious clients based on their contribution to output labels, demonstrating that it is indeed possible to defend Byzantine attacks in VFL.

## 1 Introduction

### 1.1 Background

In recent years, artificial intelligence (AI) has been applied to solve various social issues. To enhance the performance, it is necessary to train models using large amounts of data. However, when training models using vast amounts of data, data collection and the lengthy training time pose challenges in terms of computational cost. As the scale of models is expected to expand further, the issue of computational cost is an unavoidable problem. Furthermore, privacy and security issues exist, especially when handling personal information in vast amounts of data.

Federated learning (McMahan et al., 2017) is a method that reduces the computational cost for each participant (hereafter referred to as a client) by distributing model training among multiple clients while considering the privacy and security of the training data. In federated learning, each client trains a model using its dataset, and these models are integrated on a central server, resulting in a large-scale model. Each participant can ensure privacy and security by training their models while keeping their datasets confidential from the other participants. The primary federated learning methods are horizontal federated learning (HFL) (McMahan et al., 2017) and vertical federated learning (VFL) (Vepakomma et al., 2018). These training methods are classified based on the structure of each participant's dataset. While HFL has been the main focus of traditional research, advancements in VFL have been reported recently.

HFL involves training models using datasets comprising the same features and is used for tasks such as diagnosing diseases from X-ray images (Feki et al., 2021). Here, the datasets vary for clients. Each client maintains a model with the same structure, and the models are integrated. HFL requires datasets with the same features.

However, each client has its strengths and weaknesses in data collection, and there may be cases in which the collection of features is divided. In such cases, HFL, which requires each client to hold all the features, cannot be applied. VFL involves training models under the condition that each client holds a partition of the features and is used for tasks such as determining creditworthiness from transaction histories (He et al., 2023a; Luo et al., 2023). In VFL, a single model is divided among each client and the central server. This method is also referred to as split-federated learning. Only the central server knows the labels assigned to each data point. VFL involves exchanging information between each client and the central server, updating their respective models based on this information, and creating high-performance models. In the aforementioned example of creditworthiness determination, the clients are banks and stores, and the central server is the institution determining creditworthiness. This study addresses VFL.

As threats to VFL, attacks targeting both the training phase (Chen et al., 2024; Fu et al., 2022; He et al., 2023b; Xu et al., 2024; Yuan et al., 2022) and the inference phase (Duanyi et al., 2023; Pang et al., 2022) have been proposed. If the inference phase is attacked, the generated model itself remains legitimate. In contrast, if the training phase is attacked, the generated model becomes contaminated, necessitating the retraining of the model. The threat posed by attacks on the training phase is more significant, and this study focuses on discussing threats to the training phase.

The threats to VFL include attacks by malicious clients, label-inference attacks (Fu et al., 2022) and poisoning attacks (Chen et al., 2024; He et al., 2023b) including Byzantine attacks (Xu et al., 2024; Yuan et al., 2022). Label-inference attacks extract the labels of the training data held by the central server. Poison attacks degrade models by transmitting malicious data during training. This study focuses on poisoning attacks including Byzantine attacks.

Poisoning attacks are being studied actively both in terms of attack and defense methods. Poisoning attacks are assaults in which the data provider to the client or the client itself can contaminate a portion of the training data, causing the model to mispredict specific inputs without significantly reducing its accuracy. Given the vast number of data points included in a dataset, prevention through dataset verification is challenging, and various attacks (Cao & Gong, 2022; Chen et al., 2024; He et al., 2023b; Tolpegin et al., 2020) and defense methods (Cho et al., 2024; Lai et al., 2023; Lu et al., 2022; Rieger et al., 2022; Sagar et al., 2023; Xia et al., 2023) have been proposed. Notably, several existing defense methods are proposed for HFL (Lu et al., 2022; Rieger et al., 2022). In HFL, the models generated by benign clients are similar because they use the same features for training. Existing defense methods identify malicious clients by focusing on the similarity of benign cases and identifying outliers among the models sent to the central server. Furthermore, because the contaminated data are part of the training data, defense methods in VFL have been proposed that involve relocating outliers from the values sent from the client to the central server (Chen et al., 2024; Lai et al., 2023). This method combines the transmitted values from all clients and searches for outliers within the combined values, leveraging the existence of several legitimate combined values to discriminate a small number of adversarial combined values.

In VFL, the more potent Byzantine attack (Xu et al., 2024; Yuan et al., 2022) poses a realistic threat. Unlike poisoning attacks, Byzantine attacks significantly degrade the accuracy of the model by contaminating training data or tampering with the values sent to the central server. When the accuracy of the model is significantly reduced by a Byzantine attack, services utilizing that model may become dysfunctional, or substantial computational costs may be incurred for retraining the model. In HFL, the model sent from a malicious client to the central server is expected to differ significantly from those sent from other clients, making Byzantine attacks difficult to execute because of defense methods that detect outliers (Li et al., 2019; Murata et al., 2024). However, in VFL, outliers in the values transmitted from each client are meaningless because the features held by each client are inherently different. Notably, even in the absence of malicious clients, legitimate clients may be excluded. Furthermore, even if a malicious client contaminates all training data, it is challenging to detect malicious clients through outlier detection. Therefore, security evaluation of Byzantine attacks in VFL is of paramount importance.

Security assessments of Byzantine attacks on VFL have been conducted with (Xu et al., 2024) and without communication (Yuan et al., 2022) between clients, respectively. This study focuses on the latter. In the security assessment by Yuan et al., attack experiments that tampered with the values sent to the central server were conducted, and defense methods against these attacks were proposed.

Specifically, this study assumed a case in which $l_2$-regularized finite-sum minimization was solved under multiple clients, and defense was conducted using the dual space.

However, Yuan et al. presented challenges in terms of both attacks and defenses. First, regarding the attack aspect, an attack that tampers with the values sent to the central server can be detected by attaching a message authentication code (MAC) to the transmitted values. Specifically, if this process is executed in a trusted execution environment (TEE) within the client, there is no intervention by the attacker at the time of MAC attachment, making it virtually impossible to tamper with the values sent to the central server. Therefore, to appropriately assess the threat of Byzantine attacks on VFL, it is crucial to meticulously verify the attack methods under tamper detection conditions. Regarding the defense aspect, there are concerns about whether similar defenses are possible with more complex datasets and models, as the datasets and models considered are simple. The datasets they used were simplistic, consisting of linear operations with added noise, making it unclear whether their methods would be effective on real-world datasets. Moreover, Yuan et al. replaces $l_2$-regularized finite-sum minimization with a dual problem; however, when using more complex models, the dual problem may become complicated, increasing computational costs, or it may be difficult to compute the dual problem. Therefore, it is extremely important to consider simpler defense methods that do not rely on dual problems.

In this study, we improved the above two issues and conducted a more precise security assessment.

## 1.2 CONTRIBUTIONS

In this study, we investigate the feasibility of Byzantine attacks and propose new defense methods CC-VFed against these attack methods. First, in Section 3, we clarify the threat model of the attacker and classify and organize attack methods that can be executed only by polluting the training data. In particular, we qualitatively demonstrated that among the conceivable attacks, the sign-flipping attack is powerful, and we verified its threat experimentally. In Section 4, we propose a new defense method CC-VFed against Byzantine attacks using the contribution of each client to the output label. CC-VFed leverages the fact that the output labels become illegitimate in the presence of malicious clients. By removing clients that contribute significantly to the incorrectly outputted labels, it facilitates more legitimate training. In particular, unlike the prior research by Yuan et al., CC-VFed is a simpler and more practical defense method against Byzantine attacks during the training phase, which is applicable to diverse models and datasets. This study evaluates the effectiveness of CC-VFed using real-world datasets such as BCW and CIFAR10. CC-VFed serves as a defense methods against sign-flipping attacks without significantly reducing the accuracy of the original model.

## 2 PRELIMINARIES

We first introduce VFL (Vepakomma et al., 2018) in Section 2.1. In Section 2.2,, we introduce existing Byzantine attack methods (Yuan et al., 2022) against VFL.

### 2.1 VERTICAL FEDERATED LEARNING (VEPAKOMMA ET AL., 2018)

In this subsection, we discuss VFL. VFL distributes a single model among various clients and a central server. Each client and central server train their own model, thereby efficiently realizing a large-scale model.

First, we discuss the data collected by each client. In VFL, each client holds a proportion of the data with the same ID. Furthermore, only the central server holds a label corresponding to the ID. Note that each client and the central server keep the information they hold confidential from the other participants. Under these conditions, they cooperate to train a large-scale model.

Here, we explain VFL using $m$ pairs of training data and labels $(\boldsymbol{X}, Y)$. Note that $\boldsymbol{X} = [\boldsymbol{x}_1, \boldsymbol{x}_2, \ldots, \boldsymbol{x}_m]$ and $Y = [y_1, y_2, \ldots, y_m]$. This dataset is trained by $n$ clients and a central server. Client $j$ $(1 \leq j \leq n)$ holds the model $F_{\boldsymbol{\theta}_j}$ and a proportion of the divided training data $X_j = [\boldsymbol{x}_{1,j}, \boldsymbol{x}_{2,j}, \ldots, \boldsymbol{x}_{m,j}]$. Here, $\boldsymbol{\theta}_j$ is the model parameter. Additionally, by combining and appropriately rearranging the vectors $\boldsymbol{x}_{i,j}$ $(1 \leq j \leq n)$, we obtain the original data $\boldsymbol{x}_i$. The central server holds the model $F_{\boldsymbol{\theta}_0}$ and the label set $Y$. In training and inference, communication is performed between the output layer of the client and the input layer of the central server, thereby

realizing a large-scale model as a whole. To satisfy these requirements, the total number of nodes in the output layers of all the clients and the number of nodes in the input layer of the central server are the same.

Next, we discuss the specific training method. In VFL, the model parameters $\boldsymbol{\theta}_0, \boldsymbol{\theta}_1, \ldots, \boldsymbol{\theta}_n$ are updated for each data point $i = 1, 2, \ldots, m$. Although this model parameter update can be performed in batches, for simplicity, we focus on a single data $i$ and explain the model parameter update. The model parameter update is performed in the order of transmitting the output value on the client side, updating the model on the central server side, and updating the model on the client side, as follows:

1. **Output from clients:** To update the model, all clients $j$ $(1 \leq j \leq n)$ send $\boldsymbol{z}_{i,j} :=$ $F_{\boldsymbol{\theta}_j}(\boldsymbol{x}_{i,j})$ to the central server. Note that $\boldsymbol{z}_{i,j}$ is shared only between client $j$ and the central server and is kept confidential from the other clients.

2. **Renew the server model:** The central server inputs the vector $\boldsymbol{z}_i$, which is a combination of $\boldsymbol{z}_{i,j}$ from all clients, into the central server's model $F_{\boldsymbol{\theta}_0}$, and obtains the inference result $F_{\boldsymbol{\theta}_0}(\boldsymbol{z}_i)$. Then, it calculates the loss function $L_i$ using the square error or cross-entropy error, and updates the model parameter $\boldsymbol{\theta}_0$ using the error backpropagation method. Furthermore, it sends $\frac{\partial L_i}{\partial \boldsymbol{z}_{i,j}}$ to client $j$ $(1 \leq j \leq n)$.

3. **Renew client models:** Client $j$ $(1 \leq j \leq n)$ updates the model parameter $\boldsymbol{\theta}_j$ using the error backpropagation method based on the derivative of the composite function $\frac{\partial L_i}{\partial \boldsymbol{\theta}_j} = \frac{\partial L_i}{\partial \boldsymbol{z}_{i,j}} \frac{\partial \boldsymbol{z}_{i,j}}{\partial \boldsymbol{\theta}_j}$.

Thus, VFL updates the models of each client and the central server. Note that the inference is obtained by aggregating the vectors $\boldsymbol{z}_{i,j}$ output by the client's model $F_{\boldsymbol{\theta}_j}$ at the central server and then calculating $F_{\boldsymbol{\theta}_0}(\boldsymbol{z}_i)$, similar to the output of the inference result during training.

## 2.2 Previous Byzantine attacks on VFL (Yuan et al., 2022)

In this subsection, we discuss the existing Byzantine attacks on VFL conducted by Yuan et al. (Yuan et al., 2022). Yuan et al. conducted three types of attacks: Gaussian, same-value, and sign-flipping attacks based on the output vectors $\boldsymbol{z}_{i,j}$ of the other benign clients. However, in practice, information from other clients is not accessible. Therefore, in this study, instead of employing the attack methods proposed by Yuan et al., we construct a Byzantine attack by referencing the methods proposed by Ma et al. (Ma et al., 2022). Ma et al.'s attack target HFL and manipulate the local model before sending it to the central server. By drawing on these methods, a Gaussian attack, a same-value attack, and a sign-flipping attack are as follows. A Gaussian attack is an attack method in which a malicious client $j$ sends a value to the central server that adds Gaussian noise to the average of the output values of the other clients. A same-value attack is an attack method in which a malicious client sends $\boldsymbol{z}_{i,j}$, all of which have the same value, to the central server. A sign-flipping attack is an attack method in which a malicious client $j$ multiplies its output value $\boldsymbol{z}_{i,j}$ by $-c$ $(c > 1)$ and sends it to the central server. However, even if such an attack is carried out, the above attacks falsify the output value $\boldsymbol{z}_{i,j}$, and become infeasible because of tampering detection. Specifically, to bypass tampering detection, it is necessary to falsify the input data $\boldsymbol{x}_{i,j}$ and not the output value $\boldsymbol{z}_{i,j}$.

## 3 Feasible Byzantine Attacks that Tamper with Training Data

In this section, we discuss the feasibility of Byzantine attacks that tamper with training data. In Section 3.1, we discuss the attack capabilities of malicious clients. In Section 3.2 we identify the Byzantine attacks that a malicious client can execute. In particular, by comparing these attack methods, we demonstrate that the sign-flipping attack is strong in the training data. Finally, in Section 3.3, we demonstrate that the sign-flipping attack is indeed a powerful attack method.

## 3.1 The Attack Capabilities of Malicious Clients in this Paper

In this subsection, we discuss the attack capabilities of malicious clients, which are the premise of this study. Before discussing the attack capabilities, we first describe the naive defense measures that the central server can use to prevent malicious behavior by each client. These defense measures

mainly comprise encryption, tampering detection, and input/output verification and are assumed to be implemented by devices provided by the central server. In this study, we do not assume direct attacks on the central server, and the central server is assumed to operate as usual. The naive defense measures implemented by the central server are as follows:

- For each client, the model is placed within the TEE area or encrypted to prevent access to its plaintext.
- The output values of the model sent from each client to the central server are encrypted until they are received by the central server, where they are checked for tampering.
- The values sent from the central server to the client are encrypted.
- When values are input into the model, it is checked whether they fall within a specified range.

Given the above, we assume that the malicious clients considered in this study cannot perform the following:

- Obtaining training data label information (owing to vector and model encryption, label inference attacks (Fu et al., 2022) are also not possible)
- Manipulation of the model and values sent from the client to the central server
- Inputting data with out-of-range values

Based on the above, the malicious clients addressed in this paper can manipulate **only** the input to the model, specifically, the training data, and all element values must fall within the specified range. We considered strong Byzantine attacks by malicious clients with these attack capabilities.

## 3.2 CONSIDERATION OF STRONG BYZANTINE ATTACKS

In this subsection, we first classify and organize feasible Byzantine attacks, as described in Section 3.1. During this process of organization, we demonstrate the high attack capability of the sign-flipping attack. The feasible attacks can be divided into two types: attacks that set the input data $x_{i,j}$ randomly and attacks that transform it by an algorithm.

First, we consider methods that set the input data $x_{i,j}$ randomly. Because the training data comprises the data and labels, methods that manipulate either can be considered. An attack that randomly generates input data to meet the input/output conditions can be considered as a method to manipulate the data. To manipulate the labels, an attack that disrupts the correspondence between the data and labels by changing the order of the data to be input can be considered. In this type of attack, the attacker does not know the labels corresponding to the training data; therefore, it is not possible to bias the images of a specific label towards a specific class.

Next, we consider attack methods that transform input data $x_{i,j}$ using an algorithm. We consider the possibility of reproducing the attack method of Yuan et al. (Yuan et al., 2022). The consideration of more effective attack methods, particularly the optimal attack, will be a topic for future work.

First, we consider a Gaussian attack and a same-value attack. These attack methods necessitate manipulating the values sent from the client to the server; however, under the encryption of the model, it is extremely difficult to reverse calculate the model to obtain the desired output. Therefore, the Gaussian attack and the same-value attack are not feasible.

Next, we consider a sign-flipping attack. The naive sign-flipping attack manipulating the output value is not possible because of tamper detection. Here, the sign-flipping attack can be considered an input data tampering attack against the model of the central server alone. Therefore, if we consider the target of the attack to be not only the central server but also all models, including the clients, the sign-flipping attack can be established by the malicious client $j$ multiplying the input data $x_{i,j}$ $(1 \leq i \leq m)$ by $-c$ $(c > 0)$. However, if the value of $c$ is extremely small or large, there is a risk that the distribution of values may become unnatural and be detected as anomalies. Therefore, the value of $c$ needs to be set approximately close to one.

In the above discussion, we identified three types of feasible Byzantine attacks: an attack method that randomly generates training data (called "random attack"), an attack method that changes the order

of training data (called "permutation attack"), and the sign-flipping attack. Below, we qualitatively show that among these three types of attack methods, the sign-flipping attack has the highest attack effect. In particular, the sign-flipping attack shows that the loss function is significantly different from the usual function and that the gradient vector changes.

First, the sign-flipping attack changes the loss function significantly. The model comprises linear transformations and activation functions. If the sign of $x_{i,j}$ is reversed, the effect of the activation function is reversed, resulting in a significantly different value for the loss function. Note that the reversal of the effect of the activation function is more pronounced in parts that are far from the threshold, that is, parts that have significant features that make up the input.

Next, the sign-flipping attack changes the gradient vector significantly. We consider the following loss function $L\left(\boldsymbol{X}_i, \boldsymbol{\theta}\right)$, where the input $\boldsymbol{X}_i = [\boldsymbol{x}_{i,1}, \boldsymbol{x}_{i,2}, \ldots, \boldsymbol{x}_{i,n}]$ and all parameters $\boldsymbol{\theta}$ of the model are variables. Here, we consider the calculation of each element of $\frac{\partial L}{\partial \boldsymbol{\theta}}$, specifically, $\frac{\partial L}{\partial \theta_t}$ for a single parameter $\theta_t$. At this time, the loss function $L$ is expressed by the parameter $\theta_t$ to be differentiated and other model parameters in the same output layer, but these parameters are replaced by the model parameters $\boldsymbol{\theta}^*$ in the output layer before the layer where the parameter to be differentiated is located and $\boldsymbol{X}_i$. Therefore, the loss function $L$ becomes a function of $\theta_t$, $\boldsymbol{\theta}^*$, and $\boldsymbol{X}_i$, and can be expressed as a multivariate polynomial of these variables using the Maclaurin expansion. Here, we focus on the terms up to the second degree of this multivariate polynomial and ignore the remaining terms as infinitesimal. When the sign of the assigned portion $\boldsymbol{x}_{i,j}$ of client $j$ is reversed, the sign of the part assigned to $\boldsymbol{x}_{i,j}$ in the derivative of $\theta_t$ is reversed. Furthermore, the same holds for other parameters to be differentiated. In the gradient vector, (Non-derived from $\boldsymbol{x}_{i,j}$) + (Derived from $\boldsymbol{x}_{i,j}$), the sign of the second term is reversed due to the sign-flipping attack, becoming (Non-derived from $\boldsymbol{x}_{i,j}$) − (Derived from $\boldsymbol{x}_{i,j}$). Therefore, in the sign-flipping attack, the gradient vector changes, significantly disrupting the model.

Here, we apply the same discussion as that for the sign-flipping attack to the random and permutation attacks. First, not all variables' activation functions were necessarily reversed. Furthermore, the gradient vector and the overall change in the value cancel out, resulting in smaller changes compared to those in the sign-flipping attack. Therefore, it is expected that the sign-flipping attack will have the highest effect when compared with the case of randomly generating input data. Based on the above discussion, in the next subsection, we compare the random, the permutation, and the sign-flipping attack for verifying the above hypothesis.

## 3.3 COMPARATIVE EXPERIMENT OF BYZANTINE ATTACKS

This study initially conducted experiments in a scenario with two clients (referred to as Clients A and B), one of which is malicious, to validate the hypothesis posited in Section 3.2 and to ascertain the feasibility of the Byzantine attack discussed in Section 3.2. We use Ubuntu 20.04, 32GB memory, two GPUs (NVIDIA RTX A5000), using Cuda 11.6 and PyTorch 1.13.1 for Cuda 11.6. We used the numerical dataset Breast Cancer Wisconsin (BCW) (UCI Machine Learning) from UCI Machine Learning and the image dataset CIFAR10 (The Linux Foundation, a). We used the dataset included in the Python library torchvision for CIFAR10 (The Linux Foundation, b). The experiments in this study were conducted based on the implementation of Fu et al. (Fu et al., 2022). However, in this paper, to enhance the effectiveness of the defensive method proposed in Section 4, we conduct experiments using the eLU function instead of the ReLU function as the activation function for the central server. The detailed network for each dataset is shown in Appendix D, and the experimental results when using the ReLU function are presented in Appendix E. Below, we present the details of the experimental conditions for each dataset and experimental results. In this study, as in the existing research by Fu et al., we used the top-1 accuracy as the evaluation metric. Top-1 accuracy indicates the proportion of instances in the entire image dataset where the highest confidence score corresponds to the correct label. In particular, the top-1 accuracy was calculated for all test data inputs without any processing, which is different from that in the training phase.

### 3.3.1 BCW

First, we describe the experimental conditions for BCW. In BCW, binary classification was performed based on 28 numerical data points selected under the same experimental conditions as those in Fu et al. (Fu et al., 2022). Each of the 28 data points was normalized to a normal distribution

Table 1: Top-1 accuracy when conducting Byzantine attacks with two client on the BCW dataset.

| Attack type | Malicious client | |
| --- | --- | --- |
| | Client A | Client B |
| None | 93.71% | |
| Random | 97.20% | 93.71% |
| Permutation | 97.90% | 95.10% |
| Sign-flipping ($c = 1$) | 90.91% | **52.45%** |
| Sign-flipping ($c = 0.1$) | **60.84%** | **13.29%** |

with a mean of 0 and variance of 1 by StandardScaler, and then, while maintaining the order after selection, the first 14 data points were held by Client A and the last 14 data points were held by Client B. On that basis, training was conducted with 426 training data points and 143 test data. During training, the batch size was 16, number of epochs was 30, learning rate was 0.01, and error was calculated using cross-entropy. Furthermore, assuming the distribution of a slightly pre-trained model, we set the scenario such that no attack was conducted in the first epoch and attacks were conducted from the second epoch onwards.

The details of each attack method are discussed below. First, in the random attack, each element of the input was generated randomly according to a normal distribution with a mean of 0 and a variance of 1. Subsequently, during the permutation attack, the order of the input data within each batch was swapped randomly. Finally, in the sign-flipping attack, we considered two cases with $c = 1$ and $c = 0.1$. The experimental results are listed in Table 1. As shown in Table 1, although the random and permutation attacks were approximately ineffective, the sign-flipping attack significantly reduced the top-1 accuracy, corroborating the hypothesis in Section 3.2. As such, while the derivation of the optimal value of $c$ and the consideration of more powerful attack methods are future tasks, as hypothesized in Section 3.2, the sign-flipping attack possesses high attack capability.

### 3.3.2 CIFAR10

First, we describe the experimental conditions for the CIFAR10 dataset, which is an image dataset comprising 32×32 pixels. Client A held the left half of the image, and Client B held the right half. Based on this, the model structure on the client side was set to ResNet20 with ten output nodes, and training was conducted. In CIFAR10, training was conducted using 50,000 training data and 10,000 test data. During training, the batch size was 32, number of epochs was 100, learning rate was 0.1, and the error was calculated using cross-entropy. Furthermore, assuming the distribution of a slightly pre-trained model, we set the scenario such that no attack was conducted in the first five epochs, and attacks were conducted from the sixth epoch onwards.

The details of each attack method are discussed below. In the random attack, each input element was generated randomly according to a uniform distribution in the range $[0, 1)$. Subsequently, during the permutation attack, the order of the input data within each batch was swapped randomly. Finally, in the sign-flipping attack, to satisfy the input range of values, the input data were set to

$$\boldsymbol{x}'_{i,j} = [1, 1, \ldots, 1]^T - c\boldsymbol{x}_{i,j} \ (1 \leq i \leq m), \tag{1}$$

for with $c = 1$ and $c = 0.1$. The experimental results are listed in Table 2. As shown in Table 2, while the random and permutation attacks were approximately ineffective, the top-1 accuracy significantly decreases when client B conducts the sign-flipping attack with $c = 0.1$. As with BCW, the consideration of more potent attacks remains a task for future research.

## 4 DEFENSE METHODS AGAINST BYZANTINE ATTACKS

In the previous discussions, it was demonstrated that VFL can be vulnerable to Byzantine attacks. In this section, to improve upon this situation, we propose defense methods CC-VFed against Byzantine attacks at the central server. In Section 4.1, we propose the algorithm used in this study. In Section 4.2, we present the experimental results.

Table 2: Top-1 accuracy when conducting Byzantine attacks with two client on the CIFAR10 dataset.

| Attack type | Malicious client | |
|---|---|---|
| | Client A | Client B |
| None | 80.47% | |
| Random | 75.53% | 75.22% |
| Permutation | 73.64% | 74.83% |
| Sign-flipping ($c = 1$) | 74.76% | 74.97% |
| Sign-flipping ($c = 0.1$) | 74.06% | **58.82%** |

## 4.1 DEFENSE ALGORITHM

First, we describe the proposed defense algorithm CC-VFed against Byzantine attacks. CC-VFed leverages the fact that the output labels become illegitimate in the presence of malicious clients, and images are shown in Appendix A. To identify such malicious clients as described above, methods similar to Grad-CAM (Selvaraju et al., 2017) is utilized. The determination of malicious clients for one epoch was performed in the following three steps performed at the central server:

1. **Classification of raw inputs:** For one batch, the central server accepts $z_{i,j}$s from clients and outputs the label $y_i^*$s with the highest score. Furthermore, we calculate $\frac{\partial L_i}{\partial z_{i,j}}$, the gradient of the values each client sent to the central server, in order to send them to the clients determined to be malicious in Step 3.

2. **Detection of malicious clients:** For each input, if the label output $y_i^*$ in Step 1 matches the label $y_i$ of the training data, the client with a low contribution to the output label is determined to be a malicious client. Conversely, if the label output $y_i^*$ in Step 1 does not match the label $y_i$ of the training data, the client with a high contribution to the output label is determined to be a malicious client. Based on the results of the above steps performed on the input for one batch, malicious clients are identified. The method for calculating each client's contributions will be discussed in detail in Section 4.1.1, and the approach for identifying malicious clients will be elaborated in Section 4.1.2.

3. **Updating each model:** The values that malicious clients send to the central server are replaced with random values, and the central server's training is conducted while calculating the gradient for the input from each client. Then, the gradient calculated in Step 1 is then sent to the malicious clients, and the gradient calculated here is sent to the legitimate clients, updating each client's model. The method for replacing the values sent by malicious clients will be discussed in detail in Section 4.1.3.

### 4.1.1 CALCULATING EACH CLIENT'S CONTRIBUTION FOR A SINGLE INPUT

In this study, we use Grad-Cam (Selvaraju et al., 2017) to calculate the contribution of the values $z_{i,j}$ sent from each client to the output label $y_i^*$. Specifically, we set the contribution of each client $j$ as $z_{i,j} \cdot \frac{\partial F_{\theta_0}(z_i)_{y_i^*}}{\partial z_{i,j}}$. Here, $\frac{\partial F_{\theta_0}(z_i)_{y_i^*}}{\partial z_{i,j}}$ is a vector consisting of the values obtained by differentiating $F_{\theta_0}(z_i)_{y_i^*}$ with respect to each element of the vector $z_{i,j}$, and $\cdot$ represents the dot product of vectors. However, this value differs from the conventional Grad-CAM in two aspects. First, we output negative values without applying the ReLU function to the Grad-Cam values at each node to consider not only large contributions but also small contributions. Second, because the purpose of this study was to compare clients, we calculate the contribution of a client by summing up all the Grad-Cam values the client held, rather than a single node. Specifically, we calculated the contribution of each client for a single input by calculating the dot product of the vector $z_{i,j}$, which represents all the values transmitted from the client to the central server, and the vector of gradients $\frac{\partial F_{\theta_0}(z_i)_{y_i^*}}{\partial z_{i,j}}$.

### 4.1.2 IDENTIFYING MALICIOUS CLIENTS

Malicious clients are first detected for each input and then identified by aggregating these results across the batch. In this paper, we propose two methods for determining malicious clients for each input and two methods for determining malicious clients in each batch, and we propose four defense

methods by integrating them, that is., $2 \times 2 = 4$ methods. Note that the experiments conducted in this study assume a scenario in which the total number of clients is two or three and the number of malicious clients is one, as a first step to verify the effect of contributions. At most one malicious client was identified for each input and batch. However, if the model is further subdivided and the total number of clients and number of malicious clients increase, the number of identified malicious clients for each input and each batch can be set to half of the total number of clients, making the proportion of malicious clients approximately equivalent. Therefore, it is believed that the defense performance itself will also be approximately equivalent to that before subdivision. Below, we describe the method for identifying malicious clients for each input and batch.

To determine the malicious clients at each input, we made a judgment based on the comparison of the magnitude of each client's contribution, as discussed earlier. For this, the first method of judgment was a method that naively sorts the contributions of each client in ascending or descending order and identifies a specified number of malicious clients. If the number of selected malicious clients exceeded the specified number, the last added client was considered non-malicious because the contributions have the same value. However, when the output label matched the label of the training data, clients with low contributions to the output label were determined to be malicious; however, when both contributions were high, there may not be any malicious clients. Therefore, the second method of judgment is a method that sets a threshold for the contribution of clients so that it can output that there are no malicious clients. Specifically, in addition to the judgment conditions described earlier, after setting a threshold $t$, if the output label matched the label of the training data, clients with contributions of $t$ or more were considered to be non-malicious. In addition, if the output label did not match the label of the training data, clients with contributions of $t$ or less were considered to be non-malicious. In particular, we set $t = 0$ and used these two methods of judgment.

Based on the above discussion, we explain the method for determining malicious clients in each batch. The first method of judgment sorts the number of times each client has been judged to be malicious in descending order and determines a specified number of clients as malicious clients in that batch. If the number of selections exceeded the specified number for reasons such as equal numbers of malicious judgments, the last added client is considered non-malicious. As the second method of judgment, we determined the malicious clients in each batch based on the total number of times they were judged to be malicious across all batches. However, when there were no malicious clients in the relevant batch, we adhered to that judgment.

### 4.1.3 Replacing the values sent by malicious clients

From the experimental results in Section 3.3, it can be seen that the detection rate barely decreased in the random attack. Therefore, we neutralized the effects of Byzantine attacks by replacing the transmission values of clients judged as malicious with random values. The random values had the same distributions as those used for each input dataset. Specifically, in BCW, each element is randomly generated according to a normal distribution with a mean of 0 and variance of 1; and in CIFAR10, each element is randomly generated according to a uniform distribution in the range $[0, 1]$. Note that in the algorithm proposed in this paper, there may be cases where a client is mistakenly detected as malicious. Even in this case, it is believed that there will be no further attacks because the situation would be approximately the same as a normal random attack if the defense method proposed in this study is used.

### 4.2 Defense Experiments Against Byzantine Attacks

In this section, we describe defenses under the same experimental conditions as in Section 3.3 for BCW and CIFAR10 and evaluate the extent to which Byzantine attacks can be prevented. In this experiment, as stated in Section 3.3, to enhance the effectiveness of the defensive method, we conduct experiments using the eLU function instead of the ReLU function as the activation function for the central server. In Byzantine attacks, it is anticipated that the outputs of each activation function will significantly differ due to substantial alterations in information. Specifically, when using the ReLU function, nodes whose active state flips and output becomes zero will no longer be trained, thereby greatly impacting training efficiency. Therefore, in this study, we use the eLU function instead of the ReLU function to prevent the output of the activation function from becoming zero. The experimental results using the ReLU function are presented in Section Appendix E.

Table 3: Top-1 accuracy when defending against Byzantine attacks with a model of two clients on the BCW dataset. The left side of the → indicates the top-1 accuracy before defense, while the right side indicates the top-1 accuracy after defense.

| Attack type | Malicious client | |
| --- | --- | --- |
| | Client A | Client B |
| None | 93.71%→94.41% | |
| Random | 97.20%→95.10% | 93.71%→93.01% |
| Permutation | 97.90%→95.80% | 95.10%→94.41% |
| Sign-flipping ($c = 1$) | 90.91%→96.50% | **52.45%→93.71%** |
| Sign-flipping ($c = 0.1$) | **60.84%→91.61%** | **13.29%→79.02%** |

Table 4: Top-1 accuracy when defending against Byzantine attacks with a model of two clients on the CIFAR10 dataset. The left side of the → indicates the top-1 accuracy before defense, while the right side indicates the top-1 accuracy after defense.

| Attack type | Malicious client | |
| --- | --- | --- |
| | Client A | Client B |
| None | 80.47%→74.63% | |
| Random | 75.53%→75.77% | 75.22%→74.91% |
| Permutation | 73.64%→75.21% | 74.83%→75.27% |
| Sign-flipping ($c = 1$) | 74.76%→74.72% | 74.97%→75.42% |
| Sign-flipping ($c = 0.1$) | 74.06%→75.07% | **58.82%→75.52%** |

We tested all four defense methods shown in Section 3.3 and present the best experimental results here. The experimental results for all defense methods are shown in Appendix B. The experimental results for BCW are listed in Table 3, and those for CIFAR10 are listed in Table 4. From Tables 3 and 4, in addition to a significant improvement in the top-1 accuracy affected by Byzantine attacks, a high top-1 accuracy was maintained in cases unaffected by Byzantine attacks and cases without attacks. Therefore, the proposed algorithm has a significant effect as a defense method against Byzantine attacks. Furthermore, the results of experiments with three clients, as shown in Table 5, demonstrate that at least for BCW, CC-VFed against Byzantine attacks is effective. Appendix C discusses the results in detail.

## 5 CONCLUSION

In this study, we investigated the feasibility of Byzantine attacks on VFL and evaluated their safety. In Section 3, we demonstrated that the random, permutation, and sign-flipping attacks are possible attack methods that can be executed solely by pooling the training data. Among these, we qualitatively demonstrated that the sign-flipping attack is powerful and experimentally verified that the sign-flipping attack significantly reduces the performance of the model. Furthermore, in Section 4, we presented a defense method CC-VFed against Byzantine attacks using the contribution of each client to the output label, demonstrating that it serves as a defense method against sign-flipping attacks without significantly reducing the accuracy of the original model.

Table 5: Top-1 accuracy when defending against Byzantine attacks with a model of three clients on the BCW dataset. The left side of the → indicates the top-1 accuracy before defense, while the right side indicates the top-1 accuracy after defense.

| Attack type | Malicious client | | |
| --- | --- | --- | --- |
| | Client A | Client B | Client C |
| None | | 97.20%→95.10% | |
| Random | 95.80%→93.71% | 96.50%→96.50% | 96.50%→95.80% |
| Permutation | 96.50%→97.20% | 96.50%→95.80% | 97.20%→93.71% |
| Sign-flipping ($c = 1$) | 97.90%→97.90% | **89.51%→96.50%** | **50.35%→96.50%** |
| Sign-flipping ($c = 0.1$) | **54.55%→90.91%** | **41.96%→94.41%** | **41.96%→96.50%** |

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

## A    Illustrative Diagram of the Proposed Method

In this section, we present an overview of the defense method proposed in this paper. The conceptual illustration of the defense method is depicted in Figures 1 and 2. Figure 1 illustrates the scenario where the label output by the central server matches the label of the training data. In this case, a malicious client would send information that differs from the label that should be output, resulting in a smaller contribution to the output label. Conversely, Figure 2 depicts the scenario where the label output by the central server does not match the label of the training data. In this case, the malicious client would be a contributing factor to the incorrect label, leading to a larger contribution to the output label.

## B    Detailed Experimental Data for the Case of Two Clients

In this section, we present the experimental results of the four defense methods proposed in Section 4.1.2, conducted in a scenario where one of the two clients is malicious. Firstly, in Section B.1, we present the experimental results for BCW. Subsequently, in Section B.2, we display the experimental results for CIFAR10.

Before delving into each experimental result, let's first organize the defense methods. The algorithm proposed in this paper makes two types of judgments for each input and each batch. Firstly, for each

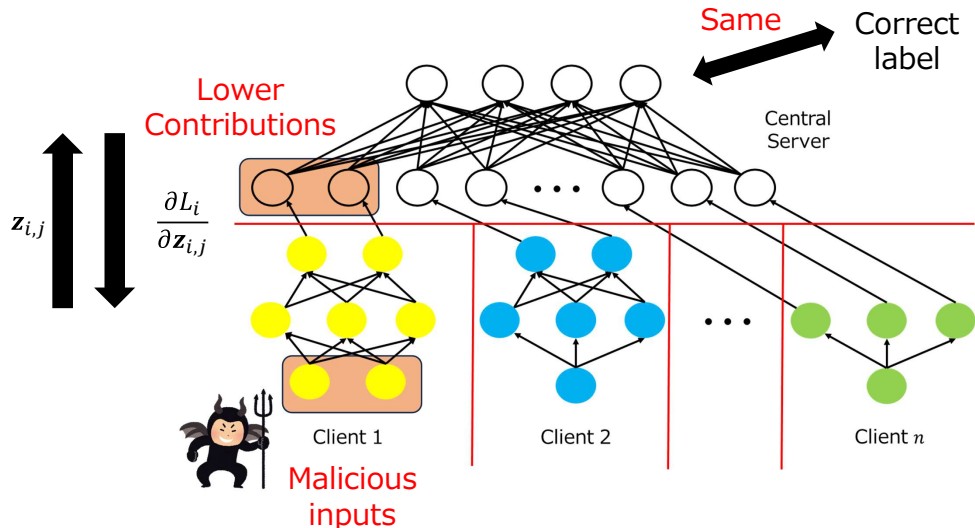

Figure 1: Illustrative diagram of the scenario when the output label is correct

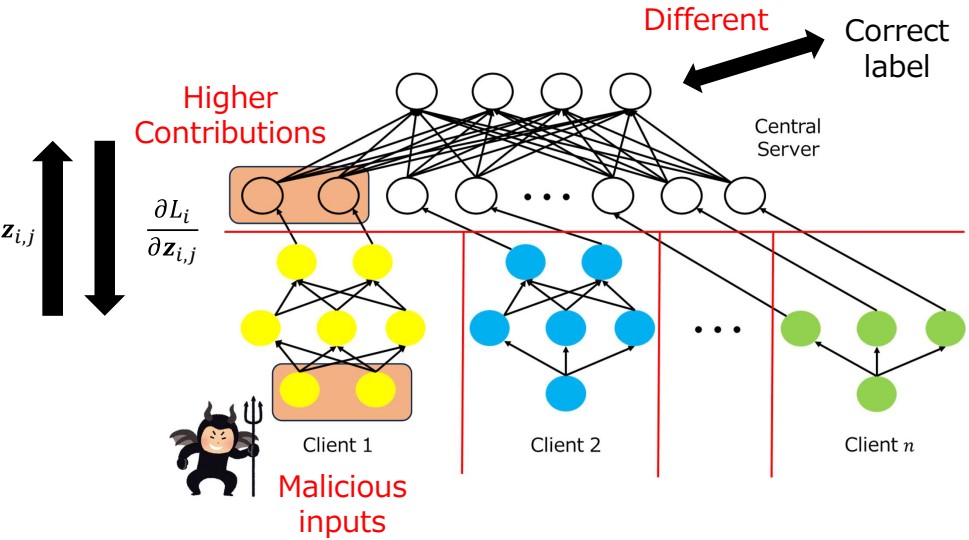

Figure 2: Illustrative diagram of the scenario when the output label is incorrect

input, we proposed two types of judgment methods based on the presence or absence of a threshold. Secondly, for each batch, we proposed two types of judgment methods based on whether or not past history is used. Below, we present the experimental results for each of these four integrated methods. To clarify each experimental condition, we will explicitly state the presence or absence of a threshold and the use or non-use of past history.

Table 6: Top-1 accuracy when defending against Byzantine attacks with a model of two clients on the BCW dataset. Threshold is used and past history is used. The left side of the $\rightarrow$ indicates the top-1 accuracy before defense, while the right side indicates the top-1 accuracy after defense.

| Attack type | Malicious client | |
|---|---|---|
| | Client A | Client B |
| None | 93.71%$\rightarrow$94.41% | |
| Random | 97.20%$\rightarrow$95.10% | 93.71%$\rightarrow$93.01% |
| Permutation | 97.90%$\rightarrow$95.80% | 95.10%$\rightarrow$94.41% |
| Sign-flipping ($c = 1$) | 90.91%$\rightarrow$96.50% | 52.45%$\rightarrow$93.71% |
| Sign-flipping ($c = 0.1$) | 60.84%$\rightarrow$91.61% | 13.29%$\rightarrow$79.02% |

Table 7: Top-1 accuracy when defending against Byzantine attacks with a model of two clients on the BCW dataset. Threshold is not used and past history is used. The left side of the $\rightarrow$ indicates the top-1 accuracy before defense, while the right side indicates the top-1 accuracy after defense.

| Attack type | Malicious client | |
|---|---|---|
| | Client A | Client B |
| None | 93.71%$\rightarrow$94.41% | |
| Random | 97.20%$\rightarrow$63.64% | 93.71%$\rightarrow$63.64% |
| Permutation | 97.90%$\rightarrow$74.13% | 95.10%$\rightarrow$63.64% |
| Sign-flipping ($c = 1$) | 90.91%$\rightarrow$11.19% | 52.45%$\rightarrow$20.28% |
| Sign-flipping ($c = 0.1$) | 60.84%$\rightarrow$13.99% | 13.29%$\rightarrow$9.79% |

## B.1 BCW

For BCW, the experimental results are shown in Tables 6–9. Here, taking the average of the top-1 accuracy after defense for the nine experimental conditions, we get 92.62% for Table 6, 46.08% for Table 7, 73.81% for Table 8, and 56.80% for Table 9. Therefore, for BCW, the defense is successful when both the threshold and past history are utilized.

Below, we discuss the reasons why the defense is successful for BCW when both the threshold and past history are utilized. The model of BCW is small and simple, and it is believed that the speed at which the model is taken over when subjected to a Byzantine attack is fast. Here, when a threshold is not used, the detection rate of malicious clients increases regardless of whether they are malicious or not, and the risk of legitimate clients being judged as malicious also increases. Furthermore, if past history is not used, there is a risk that legitimate clients may temporarily be judged as malicious. In the above scenario, if a legitimate client is judged as malicious, there will be no legitimate clients left in the training process. Consequently, in the case of BCW, where the speed at which the model is taken over is fast, the training process progresses with only the malicious client. This leads to a situation where the model trains to judge legitimate clients as malicious, further exacerbating the problem.

Table 8: Top-1 accuracy when defending against Byzantine attacks with a model of two clients on the BCW dataset. Threshold is used and past history is not used. The left side of the $\rightarrow$ indicates the top-1 accuracy before defense, while the right side indicates the top-1 accuracy after defense.

| Attack type | Malicious client | |
|---|---|---|
| | Client A | Client B |
| None | 93.71%$\rightarrow$93.01% | |
| Random | 97.20%$\rightarrow$95.10% | 93.71%$\rightarrow$95.10% |
| Permutation | 97.90%$\rightarrow$63.64% | 95.10%$\rightarrow$95.10% |
| Sign-flipping ($c = 1$) | 90.91%$\rightarrow$21.68% | 52.45%$\rightarrow$95.80% |
| Sign-flipping ($c = 0.1$) | 60.84%$\rightarrow$11.89% | 13.29%$\rightarrow$93.01% |

Table 9: Top-1 accuracy when defending against Byzantine attacks with a model of two clients on the BCW dataset. Threshold is not used and past history is not used. The left side of the $\rightarrow$ indicates the top-1 accuracy before defense, while the right side indicates the top-1 accuracy after defense.

| Attack type | Malicious client | |
| --- | --- | --- |
| | Client A | Client B |
| None | 93.71%$\rightarrow$95.80% | |
| Random | 97.20%$\rightarrow$93.71% | 93.71%$\rightarrow$69.23% |
| Permutation | 97.90%$\rightarrow$95.10% | 95.10%$\rightarrow$93.71% |
| Sign-flipping ($c=1$) | 90.91%$\rightarrow$11.19% | 52.45%$\rightarrow$18.88% |
| Sign-flipping ($c=0.1$) | 60.84%$\rightarrow$14.69% | 13.29%$\rightarrow$18.88% |

Table 10: Top-1 accuracy when defending against Byzantine attacks with a model of two clients on the CIFAR10 dataset. Threshold is used and past history is used. The left side of the $\rightarrow$ indicates the top-1 accuracy before defense, while the right side indicates the top-1 accuracy after defense.

| Attack type | Malicious client | |
| --- | --- | --- |
| | Client A | Client B |
| None | 80.47%$\rightarrow$74.73% | |
| Random | 75.53%$\rightarrow$10.00% | 75.22%$\rightarrow$10.00% |
| Permutation | 73.64%$\rightarrow$10.00% | 74.83%$\rightarrow$10.00% |
| Sign-flipping ($c=1$) | 74.76%$\rightarrow$35.61% | 74.97%$\rightarrow$74.37% |
| Sign-flipping ($c=0.1$) | 74.06%$\rightarrow$74.06% | 58.82%$\rightarrow$73.76% |

## B.2 CIFAR10

For CIFAR10, the experimental results are shown in Tables 10–13. Here, taking the average of the top-1 accuracy after defense for the nine experimental conditions, we get 41.39% for Table 10, 74.72% for Table 11, 19.14% for Table 12, and 75.17% for Table 13. Therefore, for CIFAR10, the defense is successful when threshold is not used. Below, we discuss the reasons why the defense is successful for CIFAR10 when a threshold is not utilized.

Firstly, let's focus on Table 10 and consider the reasons for the failure of the defense. In Table 10, while the defense against the sign-flipping attack is successful, the defense against the random and permutation attacks fails. CIFAR10, unlike BCW, takes time to train, and even for clean data without data contamination, the top-1 accuracy at the start of the 6th epoch is 57.08%. Therefore, compared to BCW, the probability of being judged as having an error in the output label is high. At this time, it is believed that the random and permutation attacks have a clearly smaller contribution to the output label, and legitimate clients are considered malicious. In this case, there will be no legitimate clients left in the training process, and training will proceed with only the malicious client. Conversely, in the case of random and permutation attacks, the contribution value becomes random with respect to the output label, and even if the output label is judged as legitimate when a threshold is set, there are cases where the client performing these attacks is considered non-malicious. From the above, it is believed that the risk of only legitimate clients being considered malicious is high, and the situation has become such that a model that judges legitimate clients as malicious is trained due to training only by the malicious client. It should be noted that the aforementioned phenomenon may also be attributed to the fixed threshold of zero for contributions. Therefore, setting an appropriate threshold according to the depth of training remains a task for future research.

It should be noted that the cause of failure in defending against sign-flipping attacks in Table 12 is believed to be almost the same. Sign-flipping attacks are considered to have a higher contribution to the output label compared to random or permutation attacks, as they maintain a connection with the original input. In Table 10, it is believed that the attack failed because a sufficient number of successful attacks could not be achieved, and legitimate clients could not be misjudged as malicious, resulting in the generation of a high-precision model. However, in Table 12, it is believed that even after training has progressed, legitimate clients can be misjudged as malicious without using past history, and in such cases, training progresses only with malicious clients, leading to a situation where a model that judges legitimate clients as malicious is trained.

Table 11: Top-1 accuracy when defending against Byzantine attacks with a model of two clients on the CIFAR10 dataset. Threshold is not used and past history is used. The left side of the $\rightarrow$ indicates the top-1 accuracy before defense, while the right side indicates the top-1 accuracy after defense.

| Attack type | Malicious client | |
|---|---|---|
| | Client A | Client B |
| None | 80.47%$\rightarrow$74.59% | |
| Random | 75.53%$\rightarrow$74.93% | 75.22%$\rightarrow$75.05% |
| Permutation | 73.64%$\rightarrow$74.42% | 74.83%$\rightarrow$75.06% |
| Sign-flipping ($c = 1$) | 74.76%$\rightarrow$73.57% | 74.97%$\rightarrow$74.86% |
| Sign-flipping ($c = 0.1$) | 74.06%$\rightarrow$74.98% | 58.82%$\rightarrow$75.04% |

Table 12: Top-1 accuracy when defending against Byzantine attacks with a model of two clients on the CIFAR10 dataset. Threshold is used and past history is not used. The left side of the $\rightarrow$ indicates the top-1 accuracy before defense, while the right side indicates the top-1 accuracy after defense.

| Attack type | Malicious client | |
|---|---|---|
| | Client A | Client B |
| None | 80.47%$\rightarrow$55.07% | |
| Random | 75.53%$\rightarrow$10.00% | 75.22%$\rightarrow$10.00% |
| Permutation | 73.64%$\rightarrow$10.00% | 74.83%$\rightarrow$10.00% |
| Sign-flipping ($c = 1$) | 74.76%$\rightarrow$25.77% | 74.97%$\rightarrow$25.43% |
| Sign-flipping ($c = 0.1$) | 74.06%$\rightarrow$10.00% | 58.82%$\rightarrow$15.98% |

Table 13: Top-1 accuracy when defending against Byzantine attacks with a model of two clients on the CIFAR10 dataset. Threshold is not used and past history is not used. The left side of the $\rightarrow$ indicates the top-1 accuracy before defense, while the right side indicates the top-1 accuracy after defense.

| Attack type | Malicious client | |
|---|---|---|
| | Client A | Client B |
| None | 80.47%$\rightarrow$74.63% | |
| Random | 75.53%$\rightarrow$75.77% | 75.22%$\rightarrow$74.91% |
| Permutation | 73.64%$\rightarrow$75.21% | 74.83%$\rightarrow$75.27% |
| Sign-flipping ($c = 1$) | 74.76%$\rightarrow$74.72% | 74.97%$\rightarrow$75.42% |
| Sign-flipping ($c = 0.1$) | 74.06%$\rightarrow$75.07% | 58.82%$\rightarrow$75.52% |

## C Experimental Results of Three Clients

The main experiments in this paper were conducted with two clients. In this section, we conduct experiments to see to what extent Byzantine attacks and the proposed defense method have an effect when there are three clients. Firstly, in Appendix C.1, we discuss the experimental conditions. Furthermore, in Apprndix C.2, we conduct the experiments and demonstrate that the attack and defense capabilities of Byzantine attacks heavily depend on the assigned feature quantity.

### C.1 Experimental Conditions

In this subsection, we conduct experiments in the case where there are three clients (referred to as Client A, Client B, and Client C, respectively). The dataset handled in this experiment is the same as in the previous section. Particularly, compared to Section 3 and 4, the data partitioning method and the structure of the model have changed in the experiments in this section, while all other conditions are equivalent.

Firstly, we will discuss the data partitioning method. In this experiment, for all datasets, the data is divided into three parts, held by Client A, B, and C, respectively. Firstly, in BCW, the 28 numerical data registered in the dataset are held in order, with Client A holding 9, Client B holding 9, and Client C holding 10. In CIFAR10, for the $32\times32$ pixel images, Client A holds the first 10 columns from the left, Client C holds the last 11 columns from the right, and Client B holds the remaining 11 columns in the middle. Based on this data partitioning method, the structure of the model was determined according to each dataset. The model is as shown in Tables 17 and 18 of Appendix D. It should be noted that the client model for CIFAR10 remains as ResNet20.

The above are the experimental conditions for the experiments conducted in this subsection. Here, when conducting experiments with the above partitioning method, it is expected that the contributions to training will vary greatly, and if a client who is supposed to have a high contribution is malicious, the accuracy of the model is expected to decrease significantly. For example, in image data, Clients A, B, and C are assigned the left, center, and right of the image, respectively. Since the object is likely to be in the center of the image, if Client B, who holds the center of the image, is malicious, the accuracy of the model is expected to decrease significantly, and even if other clients act maliciously, it is not expected to have much impact. Similarly, in the case of table data, it is expected that the attack results will vary depending on the data used. Also, if a client who is supposed to have a high contribution is malicious, it is expected that the defense will be less effective.

In this paper, to verify the above discussion, we trained model and calculated the top-1 accuracy for all patterns where the malicious client is Client A, Client B, or Client C. Furthermore, in this section, we utilize the defense methods CC-VFed that were shown to be highly effective for each dataset in Appendix B. Specifically, for BCW, we employ the defense method CC-VFed that uses both thresholds and historical data, while for CIFAR10, we conduct experiments under the defense method that does not use either thresholds or historical data.

### C.2 Experimental Results

The experimental results for BCW are as shown in Table 5, and those for CIFAR10 are as shown in Table 14. As can be seen from Tables 5 and 14, the attack and defend results vary depending on the part of the data used.

Firstly, we will discuss BCW. For BCW, there was not much difference in the contribution of each client, and CC-VFed was able to defend against Byzantine attacks.

Next, we will discuss CIFAR10. In CIFAR10, as hypothesized, Client B, who holds the information in the center of the image, has a high attack capability. Particularly in CIFAR10, since clients other than Client B are almost unable to attack, the information held by Client B is extremely important. Furthermore, in CIFAR10, the information occupied by Client B is large, and if Client B is malicious, the defense fails. The proposal of a defense method under this situation is a future task.

From the above, it can be concluded that in VFL, the attack and defense capabilities of Byzantine attacks greatly depend on the assigned features. As stated above, there are cases where Byzantine attacks cannot be prevented if a client with a large amount of information is malicious in the first

Table 14: Top-1 accuracy when defending against Byzantine attacks with a model of three clients on the CIFAR10 dataset. Threshold is not used and past history is not used. The left side of the $\rightarrow$ indicates the top-1 accuracy before defense, while the right side indicates the top-1 accuracy after defense.

| Attack type | Malicious client | | |
|---|---|---|---|
| | Client A | Client B | Client C |
| None | 79.42%→75.53% | | |
| Random | 77.75%→74.14% | 70.66%→37.31% | 76.59%→73.84% |
| Permutation | 77.42%→73.22% | 71.97%→26.98% | 76.56%→73.93% |
| Sign-flipping ($c = 1$) | 77.78%→71.68% | 57.55%→59.91% | 77.12%→75.13% |
| Sign-flipping ($c = 0.1$) | 78.47%→73.87% | 70.56%→45.01% | 76.55%→74.12% |

Table 15: Network for the BCW dataset (two clients)

| | Layer | #(Node) | Activation Function |
|---|---|---|---|
| | Fully connected layer | $14 \rightarrow 20$ | – |
| | Batch normalization layer | $20 \rightarrow 20$ | ReLU |
| Client | Fully connected layer | $20 \rightarrow 20$ | – |
| | Batch normalization layer | $20 \rightarrow 20$ | ReLU |
| | Fully connected layer | $20 \rightarrow 2$ | – |
| | Batch normalization layer | $4 \rightarrow 4$ | eLU |
| Server | Fully connected layer | $4 \rightarrow 4$ | – |
| | Batch normalization layer | $4 \rightarrow 4$ | eLU |
| | Fully connected layer | $4 \rightarrow 2$ | – |

place. Therefore, considering the trade-off between the efficiency of training and the amount of information assigned to each client in advance is extremely important for preventing Byzantine attacks. In particular, consideration of how to allocate data to ensure successful defense by the proposed method is a task for future research.

# D DETAILED NETWORK IN EXPERIMENTS

In this section, we provide the detailed architecture of the model network used in this study. The detailed network configurations are as shown in Tables 15–18. It should be noted that in Tables 15–18, the activation function of the central server is the eLU function, whereas in the experiments presented in Appendix E, this part is replaced with the ReLU function.

# E EXPERIMENTAL RESULTS USING RELU

In this section, we discuss the experimental results when using the ReLU function as the activation function for the central server. Firstly, in Appendix E.1, we present the experimental results for BCW. Subsequently, in Appendix E.2, we display the experimental results for CIFAR10.

Table 16: Server network for the CIFAR10 dataset (two clients)

| | Layer | #(Node) | Activation Function |
|---|---|---|---|
| | Batch normalization layer | $20 \rightarrow 20$ | eLU |
| | Fully connected layer | $20 \rightarrow 20$ | – |
| | Batch normalization layer | $20 \rightarrow 20$ | eLU |
| | Fully connected layer | $20 \rightarrow 10$ | – |
| Server | Batch normalization layer | $10 \rightarrow 10$ | eLU |
| | Fully connected layer | $10 \rightarrow 10$ | – |
| | Batch normalization layer | $10 \rightarrow 10$ | eLU |
| | Fully connected layer | $10 \rightarrow 10$ | LogSoftmax |

Table 17: Network for the BCW dataset (three clients)

|  | Layer | #(Node) | Activation Function |
|---|---|---|---|
| Client | Fully connected layer | (Input) $\to$ 14 | – |
|  | Batch normalization layer | 14 $\to$ 14 | ReLU |
|  | Fully connected layer | 14 $\to$ 14 | – |
|  | Batch normalization layer | 14 $\to$ 14 | ReLU |
|  | Fully connected layer | 14 $\to$ 2 | – |
| Server | Batch normalization layer | 6 $\to$ 6 | eLU |
|  | Fully connected layer | 6 $\to$ 6 | – |
|  | Batch normalization layer | 6 $\to$ 6 | eLU |
|  | Fully connected layer | 6 $\to$ 2 | – |

Table 18: Server network for the CIFAR10 dataset (three clients)

|  | Layer | #(Node) | Activation Function |
|---|---|---|---|
| Server | Batch normalization layer | 30 $\to$ 30 | eLU |
|  | Fully connected layer | 30 $\to$ 30 | – |
|  | Batch normalization layer | 30 $\to$ 30 | eLU |
|  | Fully connected layer | 30 $\to$ 10 | – |
|  | Batch normalization layer | 10 $\to$ 10 | eLU |
|  | Fully connected layer | 10 $\to$ 10 | – |
|  | Batch normalization layer | 10 $\to$ 10 | eLU |
|  | Fully connected layer | 10 $\to$ 10 | LogSoftmax |

### E.1 BCW

In BCW, the experimental results for the case of two clients are as shown in Tables 19–22, and for the case of three clients are as shown in Table 23. Below, we compare the eLU and ReLU functions for the defensive method that was successful in the eLU function, that is, when both the threshold and history are used. First, for the case of two clients, comparing Tables 6 and 19 reveals that the accuracy decreases when Client A performs a random attack or permutation attack. Furthermore, for the case of three clients, comparing Tables 5 and 23 shows that the defensive performance deteriorates when Client B performs a sign-flipping attack ($c = 0.1$). Therefore, using the eLU function instead of the ReLU function enhances the defensive performance of the proposed method.

### E.2 CIFAR10

In CIFAR10, the experimental results for the case of two clients are as shown in Tables 24–27, and for the case of three clients are as shown in Table 28. Below, we compare the eLU and ReLU functions for the defensive method that was successful in the eLU function, that is, when both the threshold and history are used. First, for the case of two clients, comparing Tables 13 and 27 shows that there is almost no difference, and both successfully defended against the attacks. However, for the case of three clients, comparing Table 14 and Table 28 reveals that the accuracy significantly

Table 19: Top-1 accuracy when defending against Byzantine attacks with a model of two clients on the BCW dataset. Threshold is used and past history is used. The left side of the $\to$ indicates the top-1 accuracy before defense, while the right side indicates the top-1 accuracy after defense.

| Attack type | Malicious client | |
|---|---|---|
|  | Client A | Client B |
| None | 95.10%$\to$95.80% | |
| Random | 96.50%$\to$83.92% | 95.80%$\to$93.71% |
| Permutation | 97.90%$\to$85.31% | 95.10%$\to$95.80% |
| Sign-flipping ($c = 1$) | 88.81%$\to$97.20% | 46.85%$\to$93.01% |
| Sign-flipping ($c = 0.1$) | 49.65%$\to$92.31% | 11.19%$\to$70.63% |

Table 20: Top-1 accuracy when defending against Byzantine attacks with a model of two clients on the BCW dataset. Threshold is not used and past history is used. The left side of the $\rightarrow$ indicates the top-1 accuracy before defense, while the right side indicates the top-1 accuracy after defense.

| Attack type | Malicious client | |
|---|---|---|
| | Client A | Client B |
| None | 95.10%$\rightarrow$98.60% | |
| Random | 96.50%$\rightarrow$63.64% | 95.80%$\rightarrow$83.92% |
| Permutation | 97.90%$\rightarrow$63.64% | 95.10%$\rightarrow$63.64% |
| Sign-flipping ($c = 1$) | 88.81%$\rightarrow$13.29% | 46.85%$\rightarrow$13.99% |
| Sign-flipping ($c = 0.1$) | 49.65%$\rightarrow$11.89% | 11.19%$\rightarrow$11.19% |

Table 21: Top-1 accuracy when defending against Byzantine attacks with a model of two clients on the BCW dataset. Threshold is used and past history is not used. The left side of the $\rightarrow$ indicates the top-1 accuracy before defense, while the right side indicates the top-1 accuracy after defense.

| Attack type | Malicious client | |
|---|---|---|
| | Client A | Client B |
| None | 95.10%$\rightarrow$94.41% | |
| Random | 96.50%$\rightarrow$83.92% | 95.80%$\rightarrow$63.64% |
| Permutation | 97.90%$\rightarrow$95.10% | 95.10%$\rightarrow$93.71% |
| Sign-flipping ($c = 1$) | 88.81%$\rightarrow$18.18% | 46.85%$\rightarrow$17.48% |
| Sign-flipping ($c = 0.1$) | 49.65%$\rightarrow$11.89% | 11.19%$\rightarrow$69.23% |

Table 22: Top-1 accuracy when defending against Byzantine attacks with a model of two clients on the BCW dataset. Threshold is not used and past history is not used. The left side of the $\rightarrow$ indicates the top-1 accuracy before defense, while the right side indicates the top-1 accuracy after defense.

| Attack type | Malicious client | |
|---|---|---|
| | Client A | Client B |
| None | 95.10%$\rightarrow$96.50% | |
| Random | 96.50%$\rightarrow$63.64% | 95.80%$\rightarrow$95.80% |
| Permutation | 97.90%$\rightarrow$63.64% | 95.10%$\rightarrow$65.03% |
| Sign-flipping ($c = 1$) | 88.81%$\rightarrow$9.79% | 46.85%$\rightarrow$13.29% |
| Sign-flipping ($c = 0.1$) | 49.65%$\rightarrow$10.49% | 11.19%$\rightarrow$6.99% |

Table 23: Top-1 accuracy when defending against Byzantine attacks with a model of three clients on the BCW dataset. Threshold is used and past history is used. The left side of the $\rightarrow$ indicates the top-1 accuracy before defense, while the right side indicates the top-1 accuracy after defense.

| Attack type | Malicious client | | |
|---|---|---|---|
| | Client A | Client B | Client C |
| None | 96.50%$\rightarrow$95.80% | | |
| Random | 97.20%$\rightarrow$97.20% | 96.50%$\rightarrow$96.50% | 97.20%$\rightarrow$95.80% |
| Permutation | 97.90%$\rightarrow$96.50% | 97.20%$\rightarrow$97.90% | 95.10%$\rightarrow$95.10% |
| Sign-flipping ($c = 1$) | 97.20%$\rightarrow$96.50% | 96.50%$\rightarrow$93.01% | 79.72%$\rightarrow$96.50% |
| Sign-flipping ($c = 0.1$) | 49.65%$\rightarrow$93.71% | 47.55%$\rightarrow$62.94% | 47.55%$\rightarrow$92.31% |

Table 24: Top-1 accuracy when defending against Byzantine attacks with a model of two clients on the CIFAR10 dataset. Threshold is used and past history is used. The left side of the → indicates the Top-1 accuracy before defense, while the right side indicates the top-1 accuracy after defense.

| Attack type | Malicious client | |
|---|---|---|
| | Client A | Client B |
| None | 81.01%→72.43% | |
| Random | 75.19%→10.00% | 74.95%→10.00% |
| Permutation | 74.49%→10.00% | 74.13%→10.00% |
| Sign-flipping ($c = 1$) | 74.52%→10.41% | 63.22%→72.34% |
| Sign-flipping ($c = 0.1$) | 73.95%→14.19% | 41.13%→12.62% |

Table 25: Top-1 accuracy when defending against Byzantine attacks with a model of two clients on the CIFAR10 dataset. Threshold is not used and past history is used. The left side of the → indicates the top-1 accuracy before defense, while the right side indicates the top-1 accuracy after defense.

| Attack type | Malicious client | |
|---|---|---|
| | Client A | Client B |
| None | 81.01%→73.10% | |
| Random | 75.19%→74.41% | 74.95%→75.08% |
| Permutation | 74.49%→74.05% | 74.13%→75.07% |
| Sign-flipping ($c = 1$) | 74.52%→74.99% | 63.22%→74.68% |
| Sign-flipping ($c = 0.1$) | 73.95%→74.43% | 41.13%→74.18% |

decreases when Client A or C performs a random attack or permutation attack. Therefore, using the eLU function instead of the ReLU function enhances the defensive performance of the proposed method.

Table 26: Top-1 accuracy when defending against Byzantine attacks with a model of two clients on the CIFAR10 dataset. Threshold is used and past history is not used. The left side of the → indicates the top-1 accuracy before defense, while the right side indicates the top-1 accuracy after defense.

| Attack type | Malicious client | |
|---|---|---|
| | Client A | Client B |
| None | 81.01%→30.30% | |
| Random | 75.19%→10.00% | 74.95%→10.00% |
| Permutation | 74.49%→10.00% | 74.13%→10.00% |
| Sign-flipping ($c = 1$) | 74.52%→26.79% | 63.22%→34.18% |
| Sign-flipping ($c = 0.1$) | 73.95%→10.00% | 41.13%→71.68% |

Table 27: Top-1 accuracy when defending against Byzantine attacks with a model of two clients on the CIFAR10 dataset. Threshold is not used and past history is not used. The left side of the $\rightarrow$ indicates the top-1 accuracy before defense, while the right side indicates the top-1 accuracy after defense.

| Attack type | Malicious client | |
| --- | --- | --- |
| | Client A | Client B |
| None | 81.01%$\rightarrow$75.11% | |
| Random | 75.19%$\rightarrow$74.58% | 74.95%$\rightarrow$75.01% |
| Permutation | 74.49%$\rightarrow$74.42% | 74.13%$\rightarrow$75.49% |
| Sign-flipping ($c = 1$) | 74.52%$\rightarrow$74.68% | 63.22%$\rightarrow$75.34% |
| Sign-flipping ($c = 0.1$) | 73.95%$\rightarrow$74.52% | 41.13%$\rightarrow$75.27% |

Table 28: Top-1 accuracy when defending against Byzantine attacks with a model of three clients on the CIFAR10 dataset. Threshold is not used and past history is not used. The left side of the $\rightarrow$ indicates the top-1 accuracy before defense, while the right side indicates the top-1 accuracy after defense.

| Attack type | Malicious client | | |
| --- | --- | --- | --- |
| | Client A | Client B | Client C |
| None | | 78.14%$\rightarrow$75.69% | |
| Random | 77.08%$\rightarrow$10.00% | 71.08%$\rightarrow$47.41% | 76.34%$\rightarrow$52.04% |
| Permutation | 76.73%$\rightarrow$10.00% | 69.11%$\rightarrow$45.68% | 77.43%$\rightarrow$10.00% |
| Sign-flipping ($c = 1$) | 77.16%$\rightarrow$75.64% | 52.25%$\rightarrow$58.56% | 76.71%$\rightarrow$75.50% |
| Sign-flipping ($c = 0.1$) | 77.19%$\rightarrow$72.03% | 70.39%$\rightarrow$23.77% | 76.04%$\rightarrow$73.53% |

