# OpenReview forum: "CC-VFed: Client Contribution Detects Byzantine Attacks in Vertical Federated Learning"
_ICLR.cc/2025/Conference — Submitted to ICLR 2025_

### Official Review · Reviewer_65wx · 2024-10-29

**Soundness:** 2
**Presentation:** 3
**Contribution:** 2
**Rating:** 3
**Confidence:** 4

**Summary:**

In this paper, the author proposed CC-VFed to detect and neutralize malicious clients in the setting of vertical federated learning. In general the malicious clients are recognized based on how much it contributes to mislead the VFL model into making wrong predictions. Experiments are done on 2 datasets with a limited setting in which one and only one party is malicious. The effectiveness of the proposed method is verified for defending against sing-flipping byzantine attacks for VFL.

**Strengths:**

To verify the effectiveness of the proposed defense method, a strong byzantine attack for VFL, sign-flipping attack, is first proposed and demonstrated to be quite effective in harming the performance of a VFL system. Also, studies on what kind of model architectures is more robust to attacks and helps in boosting the defense capability is also conducted which is quite interesting.

**Weaknesses:**

1. Why should $c$ be close to one? Since this argument lacks analysis or supporting work reference, it sounds really weird.
2. Experiments are not adequate, only 2 datasets are used. Besides, the setting that only one party is malicious makes the experiment and the detection of malicious party relatively easy, which could not demonstrate the effectiveness of the proposed malicious party recognition method. To what extent can this defense safeguard VFL from Byzantine attack when the number of malicious parties is unknown is not demonstrated which is a very important setting that close to real world setting.
3. I doubt the effectiveness and design of the proposed method, since when the defense is applied to defend against random/permutation attacks, the Top-1 accuracy often drop compared to without defense setting. So, the defense acks like a “stronger attack” which is not acceptable.

**Questions:**

If the authors could explain clearly why the problem I mentioned in Weakness 3 occurs and why the proposed defense is not "a stronger attack" itself, I will consider raising my score.

---

> ### Author Response · Authors · 2024-11-20
> **Response to Reviewer 65wx (Weaknesses 1)**
>
> Thanks for your valuable feedback.
>
> >Why should $c$ be close to one? Since this argument lacks analysis or supporting work reference, it sounds really weird.
>
> We agree that this argument lacks analysis or supporting work reference. The discussion regarding the value of $c$ is based on the input validation defined in this paper. During input validation, the following conditions are assumed for excluding inputs:
> - When the input values do not meet the specified upper and lower bounds.
> - When the input values significantly deviate from the expected distribution.
>
> If $c$ is significantly increased or decreased, the inputs will be excluded for the following reasons:
> - If $c$ is significantly large: The input values will not meet the specified upper and lower bounds.
> - If $c$ is significantly small: The values constituting each element will lack variance, making it easy to detect deviations from the original distribution.
>
> Thus, the inputs will be excluded based on these criteria. If the value of $c$ is close to 1, it is expected that the input will not be excluded during input validation, at least in terms of the distribution of values remaining unchanged. However, the method for setting the threshold for this input validation and the discussion regarding the permissible values of $c$ remain as topics for future research.

---

> ### Author Response · Authors · 2024-11-20
> **Response to Reviewer 65wx (Weaknesses 2)**
>
> >Experiments are not adequate, only 2 datasets are used.
>
> You have raised an important question. In this paper, we conduct experiments using the CIFAR10 image dataset and the BCW numerical dataset to verify the effectiveness of our approach on different types of data. Furthermore, the paper by Murata et al.[1], which has been accepted for ICLR 2024 and discusses defenses against Byzantine attacks in horizontal federated learning, also validates their defense methods using two image datasets, CIFAR10 and MNIST. Therefore, we consider the datasets used in our study to be sufficient.
>
> [1] Tomoya Murata, Kenta Niwa, Takumi Fukami, and Iifan Tyou. Simple minimax optimal byzantine robust algorithm for nonconvex objectives with uniform gradient heterogeneity. In The Twelfth International Conference on Learning Representations, 2024.
>
> >Besides, the setting that only one party is malicious makes the experiment and the detection of malicious party relatively easy, which could not demonstrate the effectiveness of the proposed malicious party recognition method. To what extent can this defense safeguard VFL from Byzantine attack when the number of malicious parties is unknown is not demonstrated which is a very important setting that close to real world setting.
>
> You have raised an important point. In the experiments conducted in this study, it is true that we limited the number of malicious clients to at most one. While it is indeed correct that the actual number of malicious clients is unknown and an algorithm should be constructed to accommodate this, it remains challenging to detect malicious clients with complete accuracy based on numerical calculations. Therefore, as described in Section 4.1.2, we assume that when the number of clients increases, the algorithm will detect up to half of the malicious clients. Given that we have successfully mitigated the impact of a single malicious client, it is anticipated that similar effectiveness can be achieved even if the number of malicious clients increases.

---

> ### Author Response · Authors · 2024-11-20
> **Response to Reviewer 65wx (Weaknesses 3)**
>
> >I doubt the effectiveness and design of the proposed method, since when the defense is applied to defend against random/permutation attacks, the Top-1 accuracy often drop compared to without defense setting. So, the defense acks like a “stronger attack” which is not acceptable.
>
> We agree with your assessment. In this study, while defending against sign-flipping attacks resulted in an increase in Top-1 accuracy, it is indeed possible that the Top-1 accuracy for random/permutation attacks may decrease. This can be attributed to the fact that the detection of malicious clients may occasionally be inaccurate, leading to the omission of legitimate information that is crucial for training. Specifically, in the case of random/permutation attacks, the attacks often fail to the extent that, as shown in Table 1, a more accurate model may be generated compared to the scenario where no attack is conducted. During the training process of these models, the proposed defense method in this study may inadvertently omit additional legitimate information, potentially causing a decrease in Top-1 accuracy.
> The defense method proposed in this study prioritizes algorithmic efficiency and practicality compared to the existing research by Yuan et al. However, given that the determination is made based on numerical thresholds, it is challenging to completely eliminate judgment errors, and such errors can introduce noise, slightly reducing accuracy. From this perspective, the defense method proposed in this study is challenging yet noteworthy in that it enhances accuracy through the defense mechanism while minimizing the accuracy degradation caused by judgment errors. We consider it acceptable that the Top-1 accuracy dramatically recovers in the presence of highly effective attacks, even if the detection rate decreases by a few percentage points.

---

> > ### Comment · Reviewer_65wx · 2024-11-23
> >
> > Thanks for your reply.
> >
> > From your reply, I guess the selection of the thresholds for malicious clients detection is vital for balancing between Top-1 accuracy and detection rate which is very hard to decide when no information about the attacker is known. This arises my concern about the real-world applicability of your defense mechanism. Could you provide any threshold selection cirteria to effectively select a proper threshold that fits for the potential attack?

---

> > > ### Author Response · Authors · 2024-11-26
> > > **Response to Reviewer 65wx**
> > >
> > > This is an interesting perspective. In this algorithm, two thresholds are utilized as internal parameters. Specifically, these are the threshold related to contributions, which is set to determine whether a client is malicious (as discussed in Section 4.1.2, whether to set $t=0$), and the pre-set number of malicious clients.
> > >
> > > Regarding the former threshold, which determines whether a client is malicious based on their contribution, experimental results suggest that it can be set based on the task, independent of the attack. Since the task is defined by the central server, setting a task-specific threshold (e.g., not setting $t$ for image tasks or setting $t=0$ for numerical tasks) can establish a training method capable of defending against general Byzantine attacks.
> > >
> > > As for the latter threshold concerning the number of malicious clients, it cannot be known in advance. Therefore, as discussed in Section 4.1.2 of this paper, it is set provisionally.

---

> > > > ### Comment · Reviewer_65wx · 2024-12-02
> > > >
> > > > As the hyper-parameter decision method is not well established and experimental results do reveals the importance of a proper hyper-parameter selection (if not properly selected, the defense performs like "a stronger attck" as I previously mentioned in my above comments), I decide to keep my score. Besides, I agree with reviewer t2tb that the novelty is limited.

---

### Official Review · Reviewer_VPkd · 2024-10-30

**Soundness:** 3
**Presentation:** 3
**Contribution:** 2
**Rating:** 6
**Confidence:** 3

**Summary:**

This article organizes and classifies Byzantine attacks, and proposes a new defense method CCVFed for these attack methods. Firstly, it has been proven that flipping symbol attacks pose a threat to VFL. Subsequently, in order to capture the differences in client features, this paper proposes a method for detecting and neutralizing malicious clients based on their contribution to output labels, proving that it is indeed possible to defend against Byzantine attacks in VFL.

**Strengths:**

1. The article proposes a new defense method CC-VFed, which utilizes the contribution of each client to the output label to counter Byzantine attacks.
2.   The article has verified through extensive experiments that flipping symbol attacks are threatening.
3.    In the background section, the author delves into the research question layer by layer. The outlier detection methods used in HFL scenarios are not sufficient for VFL, and there are deficiencies in the existing methods in VFL. Thus presenting the challenge of this article.
4.   This study used real-world datasets such as BCW and CIFAR10 to evaluate the effectiveness of defense methods. And the article studied the differences in using this method on different datasets.

**Weaknesses:**

1. Although the method is easy to understand, the author can better discuss the novelty of the proposed mechanism.
  2. The author should conduct a more detailed analysis of the method, identify its limitations, and make improvements.
  3. What are the unique advantages of our method compared to existing methods in other VFLs?
  4. The method lacks some key descriptions, making it difficult for people to understand

**Questions:**

1. Novelty: The novelty of the proposed method is unclear from the paper. For example, the core of the CC-VFed method mentioned in the contribution lies in the calculation of contribution degree. The author's description of two targeted improvements (Selvaraju et al., 2017) did not clearly state the effectiveness and necessity of the author's improvements. I think this needs to be strengthened. The method proposed by the author does have some practicality, but the method itself is too simple and cannot solve some key problems. For example, who will determine whether the output label matches the label of the training data? Did the author not explain this crucial issue in the methodology? If the client is to perform this critical operation, how can we prevent the client from doing evil and disrupting the correctness of the match? On the contrary, if the server executes it, then the privacy information of the data label is known by the server. How to ensure the user privacy of federated learning? These are the issues that the author needs to focus on. But I greatly appreciate the author's research on the challenges presented in this article, such as determining the most severe challenge through detailed experiments - the sign-ﬂipping attacks.
  2. Related work: What are the unique advantages of this method compared to existing methods in other VFLs? The author lacks a description and comparison of the shortcomings in other VFL methods, and does not highlight the unique advantages of the method proposed in this article, which can solve problems that cannot be solved by other methods. I think the author should add some state-of-the-art comparative literature for detailed comparison.
  3. Method: In addition to the security issues of the proposed method, some steps in the method are also confusing. For example, who will determine if the output label matches the label of the training data? The calculation formula for contribution degree lacks more description. The four defense methods proposed are actually four different situations within one defense method, which are not the four defense methods claimed by the author.
  4. Evaluation: Although the author conducted extensive experiments from various perspectives, their method did not compare with the most advanced existing methods. In addition, the author did not provide clear explanations in the experimental setup. For example, what is the experimental environment? How many experiments did the author conduct? Is the difference in results statistically significant?
  5. Organization and Writing: It is recommended that the author add some graphics to describe the proposed method, so that readers can clearly understand: which entities are there? What operations did each entity perform? What are the unique meanings behind each operation? In addition, the key standard of "Top-1 accuracy" has not been explained.

---

> ### Author Response · Authors · 2024-11-20
> **Response to Reviewer VPkd (Questions 1)**
>
> Thanks for your valuable feedback.
>
> >The novelty of the proposed method is unclear from the paper. For example, the core of the CC-VFed method mentioned in the contribution lies in the calculation of contribution degree. The author's description of two targeted improvements (Selvaraju et al., 2017) did not clearly state the effectiveness and necessity of the author's improvements. I think this needs to be strengthened.
>
> Thank you for your suggestion. In this study, we investigate the feasibility of Byzantine attacks and propose new defense methods CC-VFed against these attack methods. In particular, the novelty of this paper lies in the proposal of a new defense method against Byzantine attacks during the training phase. While Yuan et al. had previously proposed a defense method against Byzantine attacks during training, their method is only applicable to simple models, because in complex models, solving the dual problem utilized in their method may become challenging. In this study, we propose CC-VFed as a more practical defense method against Byzantine attacks during the training phase, which is applicable to diverse models and datasets. In CC-VFed, if the inference results at the central server do not match the training data, it is interpreted that a Byzantine attack has been conducted using the input data. Specifically, clients that significantly contribute to incorrect inference results are altering the inference outcomes, and thus, can be regarded as malicious clients. When detecting malicious clients, only simple computations similar to Grad-CAM are required, making this approach more practical compared to the method proposed by Yuan et al. In the revised version, we have clarified the novelty and effectiveness of our approach through this discussion.
>
> >The method proposed by the author does have some practicality, but the method itself is too simple and cannot solve some key problems. For example, who will determine whether the output label matches the label of the training data? Did the author not explain this crucial issue in the methodology? If the client is to perform this critical operation, how can we prevent the client from doing evil and disrupting the correctness of the match? On the contrary, if the server executes it, then the privacy information of the data label is known by the server. How to ensure the user privacy of federated learning? These are the issues that the author needs to focus on. But I greatly appreciate the author's research on the challenges presented in this article, such as determining the most severe challenge through detailed experiments - the sign-ﬂipping attacks.
>
> You have raised an important question. Generally, in VFL, the central server aims to output labels for a given target using data from external entities. Therefore, during the training phase, the central server retains the labels of the training data. Based on the above considerations, the comparison between the output labels and the labels of the training data is conducted by the central server.

---

> ### Author Response · Authors · 2024-11-20
> **Response to Reviewer VPkd (Questions 2)**
>
> >What are the unique advantages of this method compared to existing methods in other VFLs? The author lacks a description and comparison of the shortcomings in other VFL methods, and does not highlight the unique advantages of the method proposed in this article, which can solve problems that cannot be solved by other methods. I think the author should add some state-of-the-art comparative literature for detailed comparison.
>
> You have raised an important point. A distinctive feature of this study is its resilience to Byzantine attacks during the training phase. To the best of our knowledge, there are only two existing studies that discuss methods to defend against Byzantine attacks by clients during training: the methods proposed by Yuan et al. and Xu et al. Specifically, in vertical federated learning, the features of each client differ significantly compared to horizontal federated learning, making it difficult to detect Byzantine attacks during the training phase. This presents a particularly challenging problem. The method by Xu et al. involves communication between clients, which falls outside the scope of this study. Therefore, in this paper, we have focused on and discussed the method by Yuan et al. as a comparative benchmark. Specifically, we propose CC-VFed as a more practical defense method against Byzantine attacks during the training phase, which is applicable to diverse models and datasets. In the revised version, we have added explanations to clearly elucidate the advantages of CC-VFed.

---

> ### Author Response · Authors · 2024-11-20
> **Response to Reviewer VPkd (Questions 3)**
>
> >In addition to the security issues of the proposed method, some steps in the method are also confusing. For example, who will determine if the output label matches the label of the training data? The calculation formula for contribution degree lacks more description.
>
> We agree that some steps in the method are also confusing. The central server is responsible for making determinations regarding the output labels. Additionally, in the revised version, we have provided a more detailed description of the method for calculating contributions.
>
> >The four defense methods proposed are actually four different situations within one defense method, which are not the four defense methods claimed by the author.
>
> We agree with your assessment. Indeed, the proposed method in this study primarily utilizes contributions to detect malicious clients as a strategic approach. However, in practice, we propose four distinct methods for detecting malicious clients, recognizing that the effectiveness of each method may vary depending on the dataset. Therefore, this paper details these four methods.

---

> ### Author Response · Authors · 2024-11-20
> **Response to Reviewer VPkd (Questions 4)**
>
> >Although the author conducted extensive experiments from various perspectives, their method did not compare with the most advanced existing methods.
>
> You have raised an important point; however, we believe that there are no direct comparators for CC-VFed. Firstly, while defense methods against horizontal federated learning have been proposed, these methods are not directly applicable to VFL. Furthermore, the method proposed by Yuan et al., which is a defense method for VFL, cannot be directly applied to the datasets and models used in our experiments, making comparisons difficult. Specifically, preventing Byzantine attacks during training using practical methods is particularly challenging and has not been addressed in existing research. Therefore, we consider that there are no direct comparators for CC-VFed.
>
> >In addition, the author did not provide clear explanations in the experimental setup. For example, what is the experimental environment?
>
> We use Ubuntu 20.04, 32GB memory, two GPUs (NVIDIA RTX A5000), using Cuda 11.6 and PyTorch 1.13.1 for Cuda 11.6.
>
> >How many experiments did the author conduct? Is the difference in results statistically significant?
>
> You have raised an important question. In this study, we conducted 440 experimental patterns. Initially, for the case of 2 clients, we performed 180 experimental patterns. Specifically, as shown in Table 1 of the paper, there are 9 attack patterns, including the case where no attack is conducted. For each of these patterns, we experimented with 5 defense methods (including the case where no defense is applied), 2 datasets, and 2 activation functions, resulting in a total of 9 * 5 * 2 * 2 = 180 experimental patterns. Similarly, for the case of 3 clients, as shown in Table 5 of the paper, there are 13 attack patterns, including the case where no attack is conducted, leading to 260 experimental patterns. Summing these, we conducted a total of 440 experimental patterns. Notably, in cases where the defense is successful, the defense method succeeds against all attack patterns, thereby demonstrating the effectiveness of the defense.

---

> ### Author Response · Authors · 2024-11-20
> **Response to Reviewer VPkd (Questions 5)**
>
> >It is recommended that the author add some graphics to describe the proposed method, so that readers can clearly understand: which entities are there? What operations did each entity perform? What are the unique meanings behind each operation?
>
> Thank you for your suggestion. In the revised version, we have included figures in the Appendix A.
>
> >In addition, the key standard of "Top-1 accuracy" has not been explained.
>
> We agree with your assessment. In the revised version, we have added an explanation of Top-1 Accuracy to Section 3.3.

---

> > ### Comment · Area_Chair_NVJP · 2024-11-26
> >
> > Dear reviewer VPkd,
> >
> > Could you please respond to authors' rebuttal and see if you would like to update your review? Thanks very much!
> >
> > AC

---

### Official Review · Reviewer_t2tb · 2024-11-02

**Soundness:** 2
**Presentation:** 2
**Contribution:** 1
**Rating:** 1
**Confidence:** 4

**Summary:**

This paper compares three Byzantine attack methods in Vertical Federated Learning, demonstrating that the sign-flipping attack exhibits the highest effectiveness.  Based on the attack, they propose a new defense method CC-VFed to identify the malicious client. The proposed method's efficacy is evaluated through experiments on the BCW and CIFAR-10 datasets.

**Strengths:**

Developing Byzantine attacks in the VFL context is an important topic, and has received limited attention so far.

**Weaknesses:**

1. Limited novelty. The investigated Byzantine attack methods, particularly sign-flipping [1], have been extensively studied in previous research. The authors could have enhanced the scope by incorporating other relevant adversarial attacks in VFL [2,3]. Additionally, the gradient contribution-based defense mechanism bears significant similarities to existing approaches [4].

2. The proposed defense methodology raises concerns regarding VFL protocol compliance and privacy preservation. Specifically, Step 1 of the defense mechanism requires clients to share training data $x_i$ with the central server, which compromises data privacy. Furthermore, the aggregation process of Grad-cam values needs clarification, particularly regarding the distinction between multiple nodes versus single node scenarios. The defense computation would benefit from more rigorous mathematical formulation.

3. The experimental evaluation could be more comprehensive. The current validation relies on two relatively simple datasets (BCW and CIFAR-10), which may not fully demonstrate the method's robustness across diverse scenarios. The absence of ablation studies limits our understanding of each component's contribution to the overall system performance.

[1] Liu, Jing, et al. "CoPur: certifiably robust collaborative inference via feature purification." Advances in Neural Information Processing Systems 35 (2022): 26645-26657.
[2] Pang, Qi, et al. "ADI: Adversarial Dominating Inputs in Vertical Federated Learning Systems." arXiv preprint arXiv:2201.02775 (2022).
[3] Duanyi, Y. A. O., et al. "Constructing Adversarial Examples for Vertical Federated Learning: Optimal Client Corruption through Multi-Armed Bandit." The Twelfth International Conference on Learning Representations. 2023.
[4] J. Wang, L. Zhang, A. Li, X. You, and H. Cheng, “Efficient participant contribution evaluation for horizontal and vertical federated learning,” in 2022 IEEE 38th International Conference on Data Engineering (ICDE). IEEE, 2022, pp. 911–923.

**Questions:**

See Weakness.

---

> ### Author Response · Authors · 2024-11-20
> **Response to Reviewer t2tb (Weaknesses 1)**
>
> Thanks for your valuable feedback.
>
> >The investigated Byzantine attack methods, particularly sign-flipping [1], have been extensively studied in previous research. The authors could have enhanced the scope by incorporating other relevant adversarial attacks in VFL [2,3].
>
> You have raised an important point; however, we believe that these studies [1,2,3] would be outside the scope of our paper because these studies target the inference phase rather than the training phase. This paper targets Byzantine attacks on the training phase.
> Attacks that aim to mislead the inference results during the inference phase are indeed highly significant. However, in such attacks, the model itself has been correctly trained, so there is no need to incur substantial costs for retraining the model. In contrast, when the model itself collapses, as discussed in this paper, retraining requires significant costs, making it a more threatening attack method. Furthermore, countermeasures during the training phase are more challenging compared to those during the inference phase, as they cannot leverage the training of defense mechanisms during the training phase, as was done in the countermeasures for the inference phase [1]. Therefore, while the ultimate goal of the attacks—to mislead the detection results—remains the same, the processes involved are entirely different. In the revised version, we have clarified the above discussions.
>
> >Additionally, the gradient contribution-based defense mechanism bears significant similarities to existing approaches [4].
>
> This is a valid assessment of similarities to existing approaches [4]; however, we believe that the contributions addressed in [4] and those in this paper are different. The contribution in [4] calculates the extent to which each client's input has advanced the overall model training. In contrast, the contribution in this paper calculates the amount of information each client has provided to the output labels. While it is true that both approaches use gradients to compute some form of contribution, the targets of these computations are different. Specifically, the method proposed in this paper emphasizes the importance of evaluating changes in the output labels. The progression of model training, as indicated in literature [4], is not considered sufficient to evaluate whether an attack has occurred.

---

> > ### Comment · Reviewer_t2tb · 2024-11-27
> >
> > Thank you for your thorough revisions and clarifications.
> >
> > After careful consideration, I still have some concerns regarding the incremental nature of the work, since most pages are discussing the previous sign-flip attacks. I believe the core technical contribution could be strengthened, particularly in the algorithmic framework. The current approach, which primarily relies on gradients and labels, would benefit from a more rigorous theoretical analysis similar to that presented in your mentioned work [1]. Additionally, given the current theoretical framework, expanding the empirical validation would significantly strengthen the paper's contributions. While I appreciate authors' efforts in the revision, I maintain my previous assessment, as these aspects need further attention to fully demonstrate the paper's novelty and contribution to the field.
> >
> > [1] Tomoya Murata, Kenta Niwa, Takumi Fukami, Iifan Tyou, "Simple Minimax Optimal Byzantine Robust Algorithm for Nonconvex Objectives with Uniform Gradient Heterogeneity"

---

> > > ### Author Response · Authors · 2024-11-29
> > > **Response to Reviewer t2tb**
> > >
> > > Thank you for your comments.
> > >
> > > >After careful consideration, I still have some concerns regarding the incremental nature of the work, since most pages are discussing the previous sign-flip attacks. I believe the core technical contribution could be strengthened, particularly in the algorithmic framework.
> > >
> > > You have raised an important point; however, the discussion on the sign-flipping attack in this context aims to evaluate the effectiveness of the attack under newly imposed constraints that were not previously considered for the attacker. In this study, we have clarified that, unlike traditional scenarios, the attacker cannot freely alter the gradients when a simplified defense mechanism, different from the one proposed in this paper, is introduced. This clarification leads to a new consideration of attacks where the attacker can only manipulate the input. Therefore, in order to delineate the capabilities of the attacker, the discussion and performance evaluation of the sign-flipping attack are crucial.
> > >
> > > >The current approach, which primarily relies on gradients and labels, would benefit from a more rigorous theoretical analysis similar to that presented in your mentioned work [1]. Additionally, given the current theoretical framework, expanding the empirical validation would significantly strengthen the paper's contributions. While I appreciate authors' efforts in the revision, I maintain my previous assessment, as these aspects need further attention to fully demonstrate the paper's novelty and contribution to the field.
> > > [1] Tomoya Murata, Kenta Niwa, Takumi Fukami, Iifan Tyou, "Simple Minimax Optimal Byzantine Robust Algorithm for Nonconvex Objectives with Uniform Gradient Heterogeneity"
> > >
> > > Thank you for providing these insights. Indeed, it would be ideal to conduct a theoretical analysis of the algorithm proposed in this paper using a similar approach to [1]. In [1], the analysis of HFL leverages the high similarity of models transmitted from each client. However, in VFL, the similarity of inputs from each client is low, and the evaluation of malicious clients is based on the continuously evolving model itself due to the learning process. Therefore, conducting a similar analysis as in [1] is considered challenging. In particular, when performing the analysis, it is crucial to appropriately define assumptions regarding the attacker and the generated model. The organization of the attacker's assumptions conducted in this study is expected to significantly aid in setting these assumptions in future research.

---

> ### Author Response · Authors · 2024-11-20
> **Response to Reviewer t2tb (Weaknesses 2)**
>
> >The proposed defense methodology raises concerns regarding VFL protocol compliance and privacy preservation. Specifically, Step 1 of the defense mechanism requires clients to share training data $x_{i}$  with the central server, which compromises data privacy.
>
> We agree that the current version implies that client data is sent to the central server. This statement contains a typographical error; in reality, what is sent to the central server is not $x_{i}$ but $z_{i,j}$. In the revised version, this part has been corrected accordingly.
>
> >Furthermore, the aggregation process of Grad-cam values needs clarification, particularly regarding the distinction between multiple nodes versus single node scenarios. The defense computation would benefit from more rigorous mathematical formulation.
>
> We agree that the aggregation process of Grad-cam values needs clarification. Section 4.1.1 is structured to first present the formula for calculating each client's contribution, followed by a detailed explanation of the computation. Specifically, to calculate the contribution of a client composed of multiple nodes, the contributions of individual nodes are summed. The contribution of each node $k$ is calculated as the product of the following:
> - The input value $z_{i,j,k}$ to the server
> - The gradient of $z_{i,j,k}$ with respect to the output label
>
> Therefore, in the paper, this is expressed as the dot product of the following:
> - The input vector $z_{i,j}$ to the server
> - The gradient of $z_{i,j}$ with respect to the output label
>
> As stated initially, the calculation results are presented at the beginning of Section 4.1.1. In the revised version, we have made it clearer that these values represent the calculation results.

---

> ### Author Response · Authors · 2024-11-20
> **Response to Reviewer t2tb (Weaknesses 3)**
>
> >The experimental evaluation could be more comprehensive. The current validation relies on two relatively simple datasets (BCW and CIFAR-10), which may not fully demonstrate the method's robustness across diverse scenarios. The absence of ablation studies limits our understanding of each component's contribution to the overall system performance.
>
> You have raised an important question. In this paper, we conduct experiments using the CIFAR10 image dataset and the BCW numerical dataset to verify the effectiveness of our approach on different types of data. Furthermore, the paper by Murata et al.[5], which has been accepted for ICLR 2024 and discusses defenses against Byzantine attacks in horizontal federated learning, also validates their defense methods using two image datasets, CIFAR10 and MNIST. Therefore, we consider the datasets used in our study to be sufficient.
>
> [5] Tomoya Murata, Kenta Niwa, Takumi Fukami, and Iifan Tyou. Simple minimax optimal byzantine robust algorithm for nonconvex objectives with uniform gradient heterogeneity. In The Twelfth International Conference on Learning Representations, 2024.

---

> > ### Comment · Area_Chair_NVJP · 2024-11-26
> >
> > Dear reviewer t2tb,
> >
> > Could you please respond to authors' rebuttal and see if you would like to update your review? Thanks very much!
> >
> > AC

---

### Author Response · Authors · 2024-11-20
**Update Manuscript**

Thank you for your meaningful comments. Based on your feedback, we have made the following revisions:
- Clarified that this study targets the training phase for attacks.
- Explicitly highlighted the novelty of this research.
- Clearly compared our work with existing studies, particularly the work by Yuan et al.
- Added details regarding the experimental conditions.
- Corrected any typographical errors.
- Included illustrative diagrams of the defense mechanism in the Appendix A.

---

### Meta-Review · Area_Chair_NVJP · 2024-12-22

**Metareview:**

Strength: the paper presented a defense method against Byzantine attack in VFL.

Weakness:
1. All reviewers agreed that the paper has limited novelty. Reviewer t2tb pointed out several prior research work that was not adequately discussed and this was also stated by other two reviewers.
2. Experiments were not clearly setup and not very convincing.

**Additional Comments On Reviewer Discussion:**

Two reviewers with lower scores were actively in participating the discussions, however, neither was convinced by the authors's rebuttal. The reviewer with highest score 6 didn't participate, but the support of of this paper from this reviewer is not strong.

---

### Decision · Program_Chairs · 2025-01-22

Reject